# Bandgap Engineering via Doping Strategies for Narrowing the Bandgap below 1.2 eV in Sn/Pb Binary Perovskites: Unveiling the Role of Bi^3+^ Incorporation on Different A-Site Compositions

**DOI:** 10.3390/nano14191554

**Published:** 2024-09-26

**Authors:** Jeong-Yeon Lee, Seojun Lee, Jun Ryu, Dong-Won Kang

**Affiliations:** 1Department of Smart Cities, Chung-Ang University, 84 Heukseok-ro, Dongjak-gu, Seoul 06974, Republic of Korea; ekak123@cau.ac.kr (J.-Y.L.); sjlee6758@cau.ac.kr (S.L.); jun2019@cau.ac.kr (J.R.); 2Department of Energy Systems Engineering, Chung-Ang University, 84 Heukseok-ro, Dongjak-gu, Seoul 06974, Republic of Korea

**Keywords:** perovskite, bismuth doping, band gap narrowing, composition engineering

## Abstract

The integration of perovskite materials in solar cells has garnered significant attention due to their exceptional photovoltaic properties. However, achieving a bandgap energy below 1.2 eV remains challenging, particularly for applications requiring infrared absorption, such as sub-cells in tandem solar cells and single-junction perovskite solar cells. In this study, we employed a doping strategy to engineer the bandgap and observed that the doping effects varied depending on the A-site cation. Specifically, we investigated the impact of bismuth (Bi^3+^) incorporation into perovskites with different A-site cations, such as cesium (Cs) and methylammonium (MA). Remarkably, Bi^3+^ doping in MA-based tin-lead perovskites enabled the fabrication of ultra-narrow bandgap films (~1 eV). Comprehensive characterization, including structural, optical, and electronic analyses, was conducted to elucidate the effects of Bi doping. Notably, 8% Bi-doped Sn-Pb perovskites demonstrated infrared absorption extending up to 1360 nm, an unprecedented range for ABX_3_-type single halide perovskites. This work provides valuable insights into further narrowing the bandgap of halide perovskite materials, which is essential for their effective use in multi-junction tandem solar cell architectures.

## 1. Introduction

Recent advancements in semiconductor applications for renewable energy have attracted significant attention, particularly in the development of solar cells aimed at addressing the environmental crisis and climate change. The field of halide perovskites within solar cell technology has garnered widespread interest since its initial reports [1,2,3,4,5,6,7]. Due to their superior optoelectronic properties, bandgap tunability, and cost-effectiveness, perovskite solar cells have been considered promising alternatives to silicon-based solar cells [8,9,10,11,12,13,14,15,16,17]. Over the past decade, the power conversion efficiency (PCE) of perovskite solar cells has rapidly increased, now exceeding 26.1%, which is comparable to that of silicon solar cells [18]. Despite these attractive characteristics, challenges persist in reducing the bandgap to below 1.2 eV, a crucial requirement for their applications in photovoltaic technologies targeting the infrared (IR) spectrum. The lack of narrow-bandgap perovskites capable of absorbing longer-wavelength photons in the IR region up to 1200 nm has led to numerous attempts to achieve this reduction [19,20,21,22,23,24,25,26,27,28]. Despite various efforts, the lowest reported bandgap of an ABX_3_ perovskite device is approximately 1.2 eV, specifically for MAPb_0.2_Sn_0.8_I_3_ perovskite, based on current knowledge [29]. In response, we focused on fabricating perovskites with a bandgap narrower than 1.2 eV. In our previous work, we successfully reduced the bandgap of CsPbI_3_ inorganic perovskite by introducing Sn^2+^ [30]. Specifically, when Sn^2+^ replaced Pb^2+^ by 60%, the bandgap decreased from 1.75 eV to 1.35 eV, and phase stability improved compared to CsPbI_3_. However, the phase stability of CsSn_0.6_Pb_0.4_I_3_ (CSPI) remained inferior to that of organic cation perovskites, and the bandgap was still higher than desired. To address these challenges, we explored the addition of B-site dopants to CSPI perovskite, as doping is a common and effective strategy to modify perovskite properties. Partial substitution of B-site cations with different metal ions, such as Sr^2+^, Mn^2+^, and Co^2+^ has been proposed as a potential method to enhance the phase stability of inorganic perovskites [31,32,33]. This approach allows tuning of the perovskite structure, surface morphology, crystal size, and photoelectronic properties such as band alignment, charge transport, and defect states. In our research, we specifically focused on the effects of Bi^3+^ doping on inorganic perovskites. Bismuth (Bi^3+^) has been widely employed as a dopant in perovskite solar cells due to its similar electron configuration and ionic radius to lead (Pb^2+^) and tin (Sn^2+^), among various heterovalent ions [34,35,36,37]. Previous studies have demonstrated the potential benefits of Bi^3+^ incorporation in these materials. For example, Hu et al. reported that the incorporation of 4 mol% Bi^3+^ could stabilize the α-phase of CsPbI_3_ while simultaneously narrowing the bandgap to 1.56 eV [38]. Similarly, Kajal et al. found that a small amount of Bi^3+^ doping stabilized the distorted β-phase of CsPbI_3_ perovskite and slightly red-shifted the photoluminescence (PL) peak towards a higher wavelength [39]. Wang et al. observed that substituting Sn^2+^ with Bi^3+^ enhanced the crystallinity of CsSnI_3_ perovskite [40]. Lee et al. reported that doping 3 mol% of Bi^3+^ into CsSnI_3_ perovskite improved the stability of the B-γ phase and shifted the absorption onset to a longer wavelength [41]. These intriguing results inspired us to explore the application of Bi^3+^ doping in our research.

In this study, we incorporated Bi^3+^ into an inorganic, tin-lead (Sn-Pb) binary perovskite system, specifically focusing on CSPI, to explore the potential of Bi^3+^ doping in enhancing stability and tuning the bandgap of these materials. We further compared the effects of Bi^3+^ doping on cesium (Cs)-based and MA-based, tin-lead perovskites to understand how the A-site cation in the ABX_3_ perovskite structure influences these properties. Our results demonstrated that Bi^3+^ doping adversely impacted the crystallinity and disrupted the crystal structure of CSPI. Conversely, in MA-based, tin-lead perovskites, the crystal structure remained intact after Bi^3+^ doping, and a significant red-shift in the absorption onset was observed. Additionally, the bandgap of MA-based, tin-lead perovskites decreased progressively with increasing Bi^3+^ concentration. This study provides insight into the effects of Bi^3+^ doping on the excitonic properties, structural phases, and surface morphology of different A-site, tin-lead perovskites, highlighting the potential of Bi-doped, MA-based perovskites for extended photoresponse applications in the infrared (IR) region, up to 1360 nm.

## 2. Materials and Methods

### 2.1. Materials

Cesium iodide (CsI, 99.999%, Alfa Aesar, Incheon, Republic of Korea), Methylammonium iodide (MAI, 99.99%, Greatcell Solar, Queanbeyan, Australia) Lead(II) iodide (PbI_2_, 99.99%, Tokyo Chemical Industry Co., Ltd., Tokyo, Japan), Tin(II) iodide (SnI_2_, 99.99% Sigma-Aldrich, Saint Louis, MO, USA), tin(II) fluoride (SnF_2_, 99%, Sigma-Aldrich), were purchased and used as precursors for Sn-Pb binary perovskites. Dimethylformamide (DMF, 99.5%, Samchun Chemical, Seoul, Republic of Korea) and dimethyl sulfoxide (DMSO, 99.8%, Samchun Chemical), toluene (99.8%, Samchun Chemical), 2-Propanol (IPA, 99.99%, Sigma-Aldrich), and chlorobenzene (CBZ, 99%—GR grade, Wako, Richmond, VA, USA) were used as solvents. Acetone (≥99%, Samchun Chemical) and Isopropanol (IPA, ≥99.5%, Samchun Chemical) were used as cleaning solvents for Indium-doped tin oxide (ITO)-coated glasses (ITO, 10 Ω sq^−1^, AMG). Poly(3,4-ethylenedioxythiophene) polystyrenesulfonate (PEDOT:PSS, PVP AI 4083) was sourced from CleviosTM (Leverkusen, Germany). Phenyl-C61-butyric acid methyl ester (PC_61_BM, 99.5%) was sourced from Organic Semiconductor Materials (OSM, Seoul, Republic of Korea) and Bathocuproin (BCP, 98%) was purchased from Alfa Aesar. Silver pellet (Ag pellet, 99.99%, RND Korea, Seoul, Republic of Korea) was used as the electrode by thermal evaporator.

### 2.2. Preparation of Perovskite Precursors

A Cs-based: CsSn_0.6_Pb_0.4_I_3_ precursor was prepared from CsI, PbI_2_, SnI_2_ with 0.5 M concentration (the stoichiometric ratio of CsI:SnI_2_:PbI_2_ is 1:0.6:0.4), excess SnF_2_ of 0.03 M, as an additive. Then, BiI_3_ of and 0.005 M were added for Bi-doped Sn-Pb perovskite. After that, a DMF- and DMSO-mixed solvent (the ratio of DMF and DMSO is 0.8:0.2) was added to the prepared bottle of 1 mL. The Cs-based Sn-Pb perovskite solution was completely prepared after stirring for 2 h.

A MA-based: MASn_0.6_Pb_0.4_I_3_ precursor was prepared from MAI, PbI_2_, and SnI_2_ with 1 M concentration (the stoichiometric ratio of MAI:SnI_2_:PbI_2_ is 1:0.6:0.4), excess SnF_2_ of 0.06 M, as an additive. BiI_3_ of and 0.01, 0.02, 0.04, 0.08 M is added for Bi-doped Sn-Pb perovskite. After that, a DMF- and DMSO-mixed solvent (the ratio of DMF and DMSO is 0.8:0.2) of 1 mL was added to the prepared bottle. The MA-based, Sn-Pb perovskite solution was completely prepared after stirring for 2 h.

### 2.3. Fabrication of Tin-Lead Perovskite Thin Films and Devices

ITO-coated glasses were sequentially cleaned with acetone and IPA in the ultrasonic bath for 20 min, respectively. After cleaning, the glass substrates were dried in the dry oven at 95 °C to evaporate residual solvents. The cleaned substrates were then subjected to a UV treatment for 30 min. PEDOT:PSS was spin-coated onto the ITO substrates at 5000 rpm for 30 s in ambient conditions and annealed at 150 °C for 20 min on a hotplate as a hole transport layer (HTL) (thickness is ca. 20~30 nm). Then, the HTL-coated substrates were transferred to a N_2_-filled glove box. The Cs-based perovskite precursor solutions, with or without Bi^3+^, were spin-coated at 5000 rpm for 60 s on top of the HTL film and annealed at 80 °C for 30 min on a hot plate. The MA-based perovskite precursor solutions, with or without Bi^3+^, were spin-coated at 5000 rpm for total of 35 s (acceleration time: 5 s) in the N_2_-filled glove box. A 300 μL of toluene was applied dropwise in the center of the MA-based perovskite film at 12~13 s during the spin coating. The MA-based perovskite-coated substrate was annealed at 80 °C 10 min. For an electron transport layer, a PC_61_BM solution with 20 mg mL^−1^ dissolved in CBZ was spin-coated onto the perovskite film at 1500 rpm for 35 s and annealed at 80 °C for 10 min. A BCP solution (0.5 mg mL^−1^ dissolved in IPA) was spin-coated at 5000 rpm for 20 s and annealed at 80 °C for 5 min. The silver top electrodes (ca. 180 nm) were thermally evaporated under a high vacuum condition (<2 × 10^−6^ Torr) with a shadow mask (the size of each solar cell device is 4 mm^2^).

### 2.4. Characterizations

The optical absorption spectra were characterized using ultraviolet-visible (UV–Vis-Nir) spectroscopy (UV-2700, Shimadzu). Prior to scanning the perovskite samples, baseline correction was performed within a scan range of 300–1400 nm. The crystal structure of the perovskite films was evaluated via X-ray diffraction (XRD) (D8-Advance, Bruker-AXS). The XRD data were recorded in the two-theta range from 10° to 60°, with a scanning rate of 1° min^−1^, at room temperature (RT). The measurement conditions included a scan speed of 100 nm/min and an interval step size of 1 nm. The surface morphologies and cross-sectional images were investigated by field emission scanning electron microscopy (FE-SEM; AURIGA, Carl Zeiss). Steady-state photoluminescence (PL) spectra were acquired using a spectrofluorometer (Fluorolog3 with TCSPC, HORIBA SCIENTIFIC) with laser excitation at 463 nm.

All device characterizations were conducted under ambient conditions (RT and relative humidity of 25–30%). J-V curves of the perovskite solar cells were measured using a solar simulator (PEC-L01, Peccell Technologies) under standard AM1.5 illumination (power: 100 mW cm^−2^). EQE spectra were characterized using a CompactStat instrument (Ivium Technologies; Eindhoven, The Netherlands), which includes a power source (Abet Technologies 150 W xenon lamp, 13014) and a monochromator (DongWoo Opteron, MonoRa500i).

## 3. Results

Figure 1 presents the fundamental characterization of the absorption spectra and XRD patterns for pristine and Bi-doped CSPI perovskite films. As seen in the absorption spectra (Figure 1a), the absorbance profiles of the pristine and Bi-doped CSPI films were nearly identical. To provide greater clarity, an enlarged spectral range from 700 nm to 1200 nm is included in the inset of Figure 1a. A FE-SEM confirmed that all films had a comparable thickness of 170 ± 10 nm, ensuring that variations in film thickness did not contribute to the observed absorption differences. The absorption onset for Bi-doped samples exhibited a slight blue shift, indicating a modest increase in the bandgap. According to Tauc’s law, the absorption coefficient (α) is related to the bandgap energy (E_g_), as follows [42]:$$\alpha hv=\beta {\left(hv-E_{g}\right)}^{n}\;$$
where β is a constant independent of photon energy, and E_g_ represents the optical bandgap. The parameter n characterizes the transition process, with n = ½ for allowed direct transitions and n = 2 for allowed indirect transitions. Figure 1b and Figure 2b present the Tauc plots obtained using the above equation given direct transitions. The E_g_ value for the films was determined by extrapolating the linear portion of this relationship to its intersection with the abscissa (where α = 0, E_g_ = hν). The bandgap was found to be 1.32 eV for pristine CSPI and 1.33 eV for Bi-doped CSPI. An XRD analysis (Figure 1c,d) confirmed the crystal phase of both pristine and Bi-doped CSPI perovskites. Despite the similar absorption properties post-Bi doping, significant changes were observed in the XRD patterns.

The XRD peaks of Bi-doped CSPI were notably different from those of pristine CSPI perovskite. As shown in Appendix A, the XRD pattern for Bi-doped samples revealed several new peaks correspond to Cs_3_Bi_2_I_9_, Cs_2_SnI_6_ and tetragonal CsSnI_3_. This suggests that the incorporation of Bi^3+^ to CSPI disrupted the original structure, leading to the formation of secondary phases such as Cs_3_Bi_2_I_9_. Previous studies have reported that Cs_3_Bi_2_I_9_ has a bandgap of 1.9–2.2 eV [43,44,45], CsSnI_3_ has a bandgap of 1.85–2.49 eV [46,47], and Cs_2_SnI_6_ has a bandgap of 1.27–1.62 eV [48,49,50]. Given that the bandgaps of these decomposition phases are higher than that of CSPI, the observed bandgap increases might be due to the secondary phases. Additionally, the XRD peak intensities for Bi-doped CSPI were significantly reduced compared to those of the pristine CSPI, indicating a decrease in crystallinity.

Given that Bi-doping negatively affected the structural phase and did not reduce the bandgap of CSPI, we substituted Cs with MA at the A-site to check the effects of A-site cation in the ABX_3_ architecture on its photophysical characteristics. Interestingly, the bandgap decreased further as the MA content increased. Appendix A presents the absorption spectra of pristine and Bi-doped Cs_0.5_MA_0.5_Sn_0.6_Pb_0.4_I_3_ (Cs_0.5_MA_0.5_SPI) perovskites. For clear analysis, the graph was magnified (as shown in the inset), revealing a slight red-shift in the absorption onset towards longer wavelengths. The bandgaps were determined using the Tauc plot to be 1.23 eV and 1.2 eV for pristine and Bi-doped Cs_0.5_MA_0.5_SPI perovskites, respectively (Appendix A). To further validate the bandgap trend with increasing MA substitution for Cs, a MA-dominant composition (Cs_0.25_MA_0.75_) was also examined. Appendix A shows the absorption spectra of pristine and Bi-doped Cs_0.25_MA_0.75_Sn_0.6_Pb_0.4_I_3_ (Cs_0.25_MA_0.75_SPI) perovskites. Upon closer inspection at higher magnification, a red-shift in the bandgap toward the longer wavelength region is evident, as depicted in the inset of Figure 2c. From the Tauc plot in Appendix A, the bandgaps of pristine and Bi-doped Cs_0.25_MA_0.75_SPI were determined to be 1.2 eV and 1.14 eV, respectively. The bandgap decrement is greater in Bi-doped Cs_0.25_MA_0.75_SPI than in Bi-doped Cs_0.5_MA_0.5_SPI perovskites, suggesting that bismuth tends to act as a dopant that reduces the bandgap more effectively in MA-dominant perovskites than in Cs-based perovskites. This observation led us to further investigate the complete substitution of Cs with MA. Surprisingly, when the same characterizations were applied to pristine and Bi-doped MASn_0.6_Pb_0.4_I_3_ (MSPI), the results differed from those of CSPI perovskites. Figure 2a illustrates the absorption spectra of pristine and Bi-doped MSPI films. At a lower magnification, the graphs appeared nearly identical; however, as shown in the inset of Figure 2a, the extended spectral range from 900 nm to 1400 nm revealed a significant red-shift in the absorption onset towards longer wavelengths. The FE-SEM measurements confirmed that all films had a comparable thickness of 350 ± 10 nm, ruling out any influence of film thickness on the observed absorption differences. According to the Tauc plot analysis (Figure 2b), the bandgap was estimated to be 1.19 eV for pristine MSPI and 1.11 eV for Bi-doped MSPI. This substantial reduction in bandgap prompted further exploration by increasing Bi^3+^ concentrations. Appendix A displays the absorption spectra and Tauc plot for MSPI with varying Bi-doping ratios. Notably, the absorption onset gradually shifted towards longer wavelengths with increasing doping concentration, as shown in Appendix A. A clear reduction in the bandgap was observed from the Tauc plot (Appendix A), with the bandgap reaching 1.055 eV for 8% Bi-doped MSPI. This significant extension of the absorption range following Bi-doping is consistent with reports on other MA-based perovskites such as MAPbCl_3_, MAPbBr_3_, and MAPbI_3_. Zhang et al. observed a considerable red-shift in the absorption onset with increasing Bi^3+^ content in MAPbCl_3_ single crystals [51]. The minimum bandgap of 2.62 eV was obtained for Bi-doped MAPbCl_3_, which is 300 meV narrower than that of pristine MAPbCl_3_ (2.92 eV). By combining theoretical calculations with experimental results, they speculated that the energy level of the empty 6p orbitals of Bi^3+^ is lower than that of Pb^2+^. Thus, with increasing Bi^3+^ content, the conduction band minimum (CBM) shifts downward, narrowing the bandgap. Similarly, Abdelhady et al. found that the absorption spectra displayed a red-shift proportional to the Bi^3+^ content in MAPbBr_3_ crystals [52]. Based on density functional theory (DFT) calculations, they suggested that the relatively shallow and delocalized Bi^3+^ states facilitate interactions between nearby Bi^3+^ dopants, creating conditions conducive to bandgap narrowing. Likewise, Wang et al. used Bi^3+^ as a dopant in MAPbI_3_ perovskite and observed a red-shift of 140 meV in the absorption onset [53]. They proposed that the bandgap reduction could be attributed to the creation of impurity bands within the bandgap. Due to their similar electron configurations and ionic radii, Bi^3+^ ions can be incorporated into the crystal lattice of lead perovskites, generating Bi_Pb_ defects that act as shallow donors and introduce energy levels below the CBM. Therefore, we proceeded with the fabrication of devices for Bi-doped MAPbI_3_ perovskites to verify the bandgap change, as detailed in the following sections. The crystal phases of pristine and Bi-doped MAPbI_3_ perovskites were confirmed by XRD measurements. The XRD pattern (Figure 2c,d) shows the characteristic tetragonal phase structure at room temperature, with main diffraction peaks at 2θ values of 14° and 28°. The diffraction pattern for Bi-doped perovskite films did not reveal any new peaks and was largely consistent with that of pristine MSPI perovskites. Although the peak intensity for Bi-doped samples was lower than that of the pristine samples, the crystal phase was better preserved compared to Bi-doped CSPI perovskite. Table 1 compares the band gap and crystallographic parameters of CSPI perovskite films with those of MSPI perovskite films.

Figure 3 presents the surface FE-SEM images of pristine and Bi-doped CSPI and MSPI perovskite thin films. As shown in Figure 3a,b, the Bi-doped CSPI perovskite thin film exhibits more pinholes and a smaller grain size compared to the pristine film. In contrast, the surface morphology of the Bi-doped MSPI perovskite thin film (Figure 3c,d) closely resembles that of the pristine film. The pristine MSPI perovskite thin film displays distinct grain domains of approximately 300 nm and a pinhole-free, homogeneous surface. The Bi-doped MSPI perovskite thin film also shows a pinhole-free surface, albeit with slightly smaller and less uniform grain sizes compared to the pristine film. Despite the somewhat degraded surface quality of the Bi-doped MSPI perovskite thin film, the overall morphology remains relatively uniform and compact. These contrasting outcomes align with the structural analysis trends. As previously mentioned, the inherent phase instability of CSPI, due to its small tolerance factor (Appendix A), prevents it from maintaining its original structure. The XRD measurements revealed that Bi-doping led to the decomposition of the CSPI structure into several phases. Consequently, Bi incorporation into the CSPI perovskite negatively impacted crystal growth, which is reflected in the morphological properties. After SEM analyses, energy dispersive X-ray spectroscopy (EDS) was performed to determine the elemental composition of pristine and 8% Bi-doped MSPI perovskites (Appendix A). The specific atomic percentages and the Sn/Pb ratio are provided in the inset table. Since tin(II) fluoride (SnF_2_) was added to suppress the oxidation of Sn^2+^ to Sn^4+^, and due to its surface accumulation properties, the Sn content might be detected to be over 60% in all samples. Additionally, the Bi content in the thin films of Bi-doped MSPI perovskite was lower than the nominal Bi content in the precursor solutions, consistent with previous reports on Bi-doped FAPbI_3_ [37]. According to the EDS results, the Sn/Pb ratio decreased from 1.92 to 1.83 after Bi-incorporation, indicating that Sn might be mainly replaced by Bi. Prior reports, supported by density functional theory (DFT) calculations, have shown that the formation energy of Bi_Sn_ is lower than that of Bi_Pb_, suggesting a higher likelihood of Sn substitution by Bi compared to Pb. Furthermore, the ionic radius of Bi^3+^ (1.03 Å) is closer to that of Sn^2+^ (1.04 Å) than to Pb^2+^ (1.19 Å) [54], making Sn replacement by Bi more probable in this structure.

Given that surface quality significantly influences device performance, only MA-based perovskites were further developed for device fabrication.

To assess the impact of Bi-doping on solar cell performance, devices were fabricated using pristine and Bi-doped MSPI perovskites as the light-absorbing layer. The preparation process for these devices is depicted in Figure 4a. The J-V curves (Appendix A) for both device types indicate a significant degradation in the photovoltaic performance of Bi-doped MSPI perovskite devices. A notable decrease in open-circuit voltage (V_oc_) was observed, corresponding to a sharp reduction in PL intensity (Appendix A). This substantial PL quenching was also observed in Bi-doped MAPbI_3_ and CsSnI_3_ perovskites [41,53]. It was confirmed that Bi^3+^ incorporation generates Bi_Pb_ defects, which act as shallow donors and introduce energy levels below the conduction band minimum (CBM) in MAPbI_3_ perovskites. Similarly, the decrease in PL intensity of Bi-doped CsSnI_3_ perovskites can be attributed to the formation of intermediate bands below the conduction band due to the BiI_3_ doping. Therefore, the PL decrease can be explained by the enhanced non-radiative recombination, as carriers are trapped in the intermediate energy bands introduced by bismuth. Moreover, the V_oc_ loss is also likely due to the defect states introduced by Bi^3+^ and reduced grain size and increased grain boundaries within the bulk material. Additionally, the short-circuit current density (J_sc_) exhibited a more pronounced decline than V_oc_, dropping from 24.73 mA cm^−2^ to approximately 0.3–0.4 mA cm^−2^ after Bi-doping. This drastic reduction in J_sc_ is consistent with the earlier explanation for the bandgap decrease. Although light absorption is a key factor influencing short-circuit current, UV-Vis measurements demonstrated that increased Bi concentration did not affect light absorption. The loss in J_sc_ may instead be attributed to increased bulk recombination, leading to reduced charge extraction. This increase in bulk recombination could result from strong doping of the perovskite bulk or the presence of space charges caused by filled traps or accumulated ionic species. Furthermore, as indicated by the FE-SEM images, the deteriorated surface morphology and increased grain boundaries in the Bi-doped MSPI thin film could serve as trap sites, facilitating charge capture and further reducing short-circuit currents. Interestingly, a progressive decline in performance was observed in proportion to the Bi^3+^ concentration, as shown in Figure 4b. The photovoltaic performance factors are summarized in Table 2.

To further understand the effect of Bi-doping on charge carrier diffusion, the EQE of the devices was measured, as shown in Figure 4c,d. From the normalized EQE, it was observed that the NIR charge collection was significantly enhanced when Bi was incorporated. This result consistent with the absorption measurements in Figure 2a and Appendix A. As shown in Appendix A, the EQE sharply decreased for Bi-doped devices, confirming the poor performance previously noted. This suggests that Bi^3+^ incorporation creates trap sites in the thin film, reducing carrier collection efficiency. Additionally, this performance degradation could stem from reduced charge carrier mobility, caused by increased asymmetric carrier trapping at grain boundaries and shorter charge carrier lifetimes due to diminished intrinsic electronic quality. Nonetheless, a clear red-shift in the onset of the normalized EQE spectrum was observed following Bi-doping. Figure 4c illustrates that the Bi-doped solar cells collected longer-wavelength spectra more effectively than the pristine cells, extending the device response from 1060 nm in the pristine state to 1260 nm in cells doped with 1% Bi. As shown in Figure 4d, the EQE response continued to extend with increasing Bi concentration, reaching an operational wavelength of up to 1360 nm in 8% Bi-doped MSPI. This unprecedented extension of the operational range confirms that Bi-doping enhances the photoresponse of MSPI perovskites, as evidenced by both absorption and EQE measurements. It is deemed necessary to address the technical issues related to the defects currently observed by developing additional compositions and additive processes. Nevertheless, this study has identified the lowest bandgap (~1360 nm, 1.055 eV) that can be achieved in a single halide perovskite with an ABX_3_ structure, which is quite significant. With further research, new materials that strongly absorb near-infrared bands and suppress defect formation could dominate next-generation perovskite technology and potentially challenge solar cells based on III-V compounds.

## 4. Conclusions

This study presents the first development of Bi-doped CSPI tin-lead binary perovskite, inspired by prior reports indicating enhanced phase stability and reduced bandgap in Bi-doped unary perovskites. Contrary to expectations, Bi^3+^ incorporation into CSPI adversely affected the structure, leading to the formation of secondary decomposition phases and an unexpected increase in the bandgap. Additionally, surface morphology deteriorated. However, when the A-site cation was switched from Cs to MA, significant improvements were observed, highlighting the critical importance of A-site selection in doping strategies. While Bi^3+^ incorporation led to some deterioration in photovoltaic properties, it successfully reduced the bandgap of MSPI perovskite by more than 0.1 eV, achieving an unprecedented narrow bandgap of 1.055 eV in solution-processed halide perovskite films. This is a significant advancement, as current research on perovskites with bandgaps below 1.2 eV is limited. Despite performance challenges, Bi^3+^ doping remains a promising approach for developing super-narrow bandgap perovskite solar cells with extended infrared photo-response up to ~1400 nm. Further investigation into the underlying mechanisms of Bi^3+^ incorporation based on A-site composition will be crucial for advancing bandgap engineering strategies in ultra-narrow bandgap perovskites.

## Figures and Tables

**Figure 1 nanomaterials-14-01554-f001:**
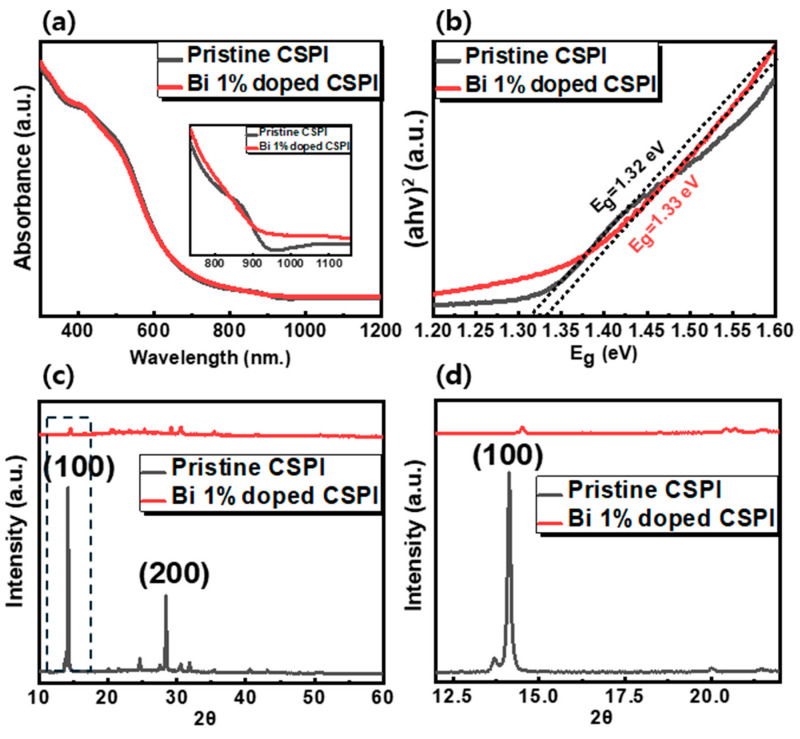
(**a**) Absorption spectra of pristine and Bi 1% doped CSPI perovskite films, with the inset highlighting the enlarged absorption spectrum, (**b**) Tauc plot of pristine and Bi 1% doped CSPI perovskite films, (**c**) Stacked XRD patterns of pristine and Bi 1% doped CSPI perovskite films with dashed box highlighting the main peak, (**d**) Main XRD peak of pristine and Bi 1% doped CSPI perovskite films.

**Figure 2 nanomaterials-14-01554-f002:**
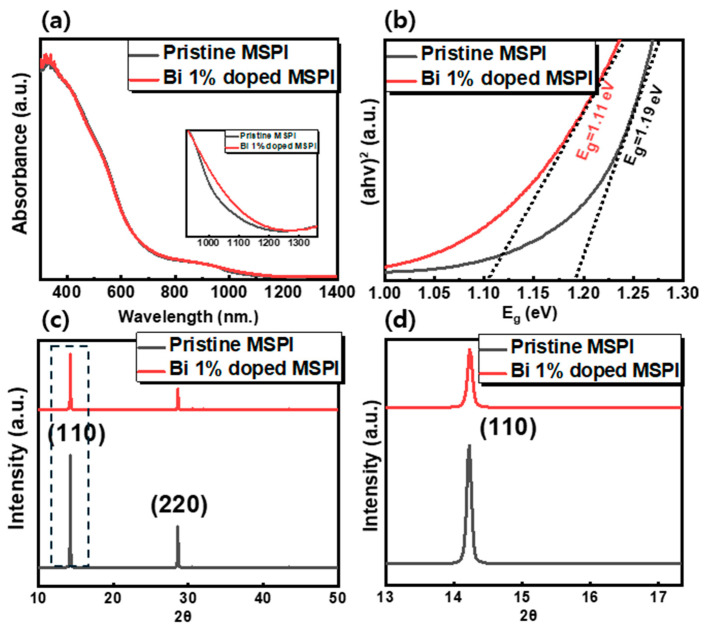
(**a**) Absorption spectra of pristine and Bi 1% doped MSPI perovskite films, with the inset highlighting the enlarged absorption spectrum, (**b**) Tauc plot of pristine and Bi 1% doped MSPI perovskite films, (**c**) Stacked XRD patterns of pristine and Bi 1% doped MSPI perovskite films with dashed box highlighting the main peak, (**d**) Main XRD peak of pristine and Bi 1% doped MSPI perovskite films.

**Figure 3 nanomaterials-14-01554-f003:**
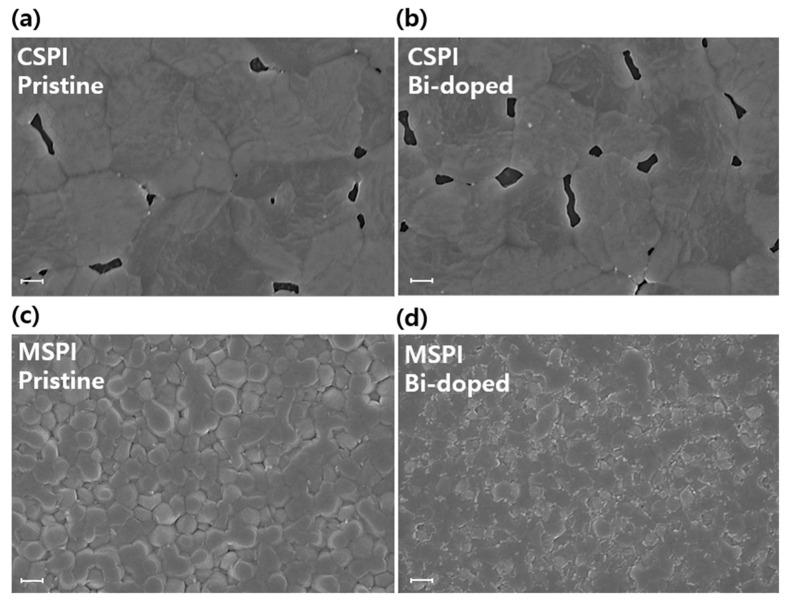
(**a**) Surface FE-SEM image of pristine CSPI perovskite film on a glass substrate, (**b**) Surface FE-SEM image of Bi-doped CSPI perovskite film on a glass substrate, (**c**) Surface FE-SEM image of pristine MSPI perovskite film on a glass substrate, (**d**) Surface FE-SEM image of Bi-doped MSPI perovskite film on a glass substrate. Scale bar: 300 nm.

**Figure 4 nanomaterials-14-01554-f004:**
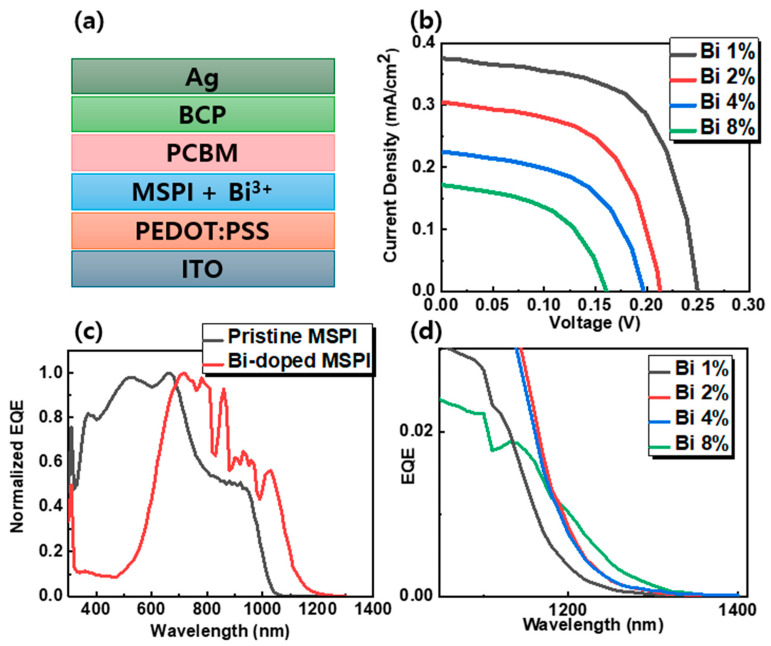
(**a**) Device configuration diagram, (**b**) J-V curves of 1%, 2%, 4%, and 8% Bi-doped MSPI devices, (**c**) Normalized EQE spectrum of pristine and Bi 1% doped MA-based devices, (**d**) Enlarged EQE spectrum of 1%, 2%, 4%, and 8% Bi-doped MSPI devices.

**Table 1 nanomaterials-14-01554-t001:** Comparison of bandgap and crystallographic parameters for pristine and Bi-doped CSPI and MSPI perovskite films.

	Bi^3+^ Concentration [%]	Band Gap [eV]	Lattice Parameter [Å]	Crystallite Size [nm]
CSPI	0	1.32	6.27	78.34
	1	1.33	6.10	66.38
MSPI	0	1.19	6.21	86.08
	1	1.11	6.22	77.13

**Table 2 nanomaterials-14-01554-t002:** Photovoltaic Performance Factors for Pristine and Bi-doped MSPI Perovskite Devices.

Bi Concentration [%]	V_OC_ [V]	J_SC_ [mA cm^−2^]	FF [%]	PCE [%]
0	0.656	24.73	73.58	11.95
1	0.2501	0.375	62.0258	0.0582
2	0.2122	0.3	60.07	0.0383
4	0.197	0.225	57.24	0.0254
8	0.155	0.175	50.69	0.0138

## Data Availability

The original contributions presented in the study are included in the article/Appendix A, further inquiries can be directed to the corresponding author.

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
