# Peer review of "Bandgap Engineering via Doping Strategies for Narrowing the Bandgap below 1.2 eV in Sn/Pb Binary Perovskites: Unveiling the Role of Bi3+ Incorporation on Different A-Site Compositions"

_nanomaterials, 2024, doi:10.3390/nano14191554_

Round 1

Reviewer 1 Report

Comments and Suggestions for Authors

A new bandgap engineering on Bi3+ doped unary/binary perovskite is reported. In particularly, the role of Bi3+ on the Cs/MA-based perovskites is analyzed in detail. It is found that the effect of Bi3+ on the MA-based perovskite is superior to the others. So I think that the manuscript can be acceptable in publication in NANOMATERIALS after the following revisions.

1.       The manuscript is on the role of Bi3+ incorporation on the different A-site ions. However, there is only the role for unary perovskites. What about the role for binary perovskites?

2.       As well known, the bandgap can be gained by absorption spectra. However, the bandgaps are herein obtained by the sub-plots Fig.1(b) and Fig.2(b). What is the difference? More instructions should be delivered.

3.       The normalized EQE should be elucidated appropriately in the text.

Comments on the Quality of English Language

Minor editing of English language required.

Author Response

Please kindly check the attached response letter below.

Reviewer 2 Report

Comments and Suggestions for Authors

This paper on Bi doped perovskite is of real interest for the PV community.

However, some additionnal discussion are needed before acceptance:

- which metal (Pb or Sn) is mainly replaced by Bi?

- what can explain the PL decrease avec Bi doping?

Other minor corrections are also listed below:

- Fig S1 : 2 legends (repetition) (+ no figure b)

- Table S1: the legend is not corresponding to the table

- what is the tolerance factor of Table S1 -> way of calcul?

- Lines 103 and 108: SnF2 is mentioned in the text (I think it should be SnI2)

- Please precise the % of Bi doping in the legend of figures 1 and 2

Author Response

(The authors gave the same response as above.)

Round 2

Reviewer 2 Report

Comments and Suggestions for Authors

The EDS results presented in the answer to reviewers should also be included in the SI file -> their give a very pertinent information

Author Response

Comment 1: "The EDS results presented in the answer to reviewers should also be included in the SI file -> their give a very pertinent information"

=>

Reply 1: Thank you so much for the careful comment to improve our work. 

As the reviewer pointed out, we have included the EDS results to SI file. 

Thank you so much!